# Clinical Evaluation of the ButterfLife Device for Simultaneous Multiparameter Telemonitoring in Hospital and Home Settings

**DOI:** 10.3390/diagnostics12123115

**Published:** 2022-12-10

**Authors:** Francesco Salton, Stefano Kette, Paola Confalonieri, Sergio Fonda, Selene Lerda, Michael Hughes, Marco Confalonieri, Barbara Ruaro

**Affiliations:** 1Pulmonology Unit, Department of Medical Surgical and Health Sciences, University Hospital of Cattinara, University of Trieste, 34149 Trieste, Italy; 2Department of Biomedical Sciences, University of Modena and Reggio Emilia, 41121 Modena, Italy; 324ORE Business School, Via Monte Rosa, 91, 20149 Milan, Italy; 4Division of Musculoskeletal and Dermatological Sciences, Faculty of Biology, Medicine and Health, The University of Manchester & Salford Royal NHS Foundation Trust, Manchester M6 8HD, UK

**Keywords:** ButterfLife, telehealth, telemedicine, vital parameters, ECG, photoplethysmography, artificial intelligence

## Abstract

We conducted a two-phase study to test the reliability and usability of an all-in-one artificial intelligence-based device (ButterfLife), which allows simultaneous monitoring of five vital signs. The first phase of the study aimed to test the agreement between measurements performed with ButterfLife vs. standard of care (SoC) in 42 hospitalized patients affected by acute respiratory failure. In this setting, the greatest discordance between ButterfLife and SoC was in respiratory rate (mean difference −4.69 bpm). Significantly close correlations were observed for all parameters except diastolic blood pressure and oxygen saturation (Spearman’s Rho −0.18 mmHg; *p* = 0.33 and 0.20%; *p* = 0.24, respectively). The second phase of the study was conducted on eight poly-comorbid patients using ButterfLife at home, to evaluate the number of clinical conditions detected, as well as the patients’ compliance and satisfaction. The average proportion of performed tests compared with the scheduled number was 67.4%, and no patients reported difficulties with use. Seven conditions requiring medical attention were identified, with a sensitivity of 100% and specificity of 88.9%. The median patient satisfaction was 9.5/10. In conclusion, ButterfLife proved to be a reliable and easy-to-use device, capable of simultaneously assessing five vital signs in both hospital and home settings.

## 1. Introduction

The growing prevalence and cost of chronic diseases are an increasing burden on health care systems. For more than two decades, telemedicine has proven effective in improving patients’ quality of life and increasing the efficiency of care processes [1]. Telemedicine is the remote delivery of healthcare services, through electronic communication, that allows healthcare providers to evaluate, diagnose and treat patients without the need for an in-person visit [2]. Beginning more than 40 years ago with hospitals extending their services to patients in remote locations, telemedicine has grown rapidly and has become an integrated part of specialty departments, hospitals, private doctor offices, and home health care [3]. Telemedicine applications include, among others, (a) virtual care visits, (b) telemonitoring, (c) telerehabilitation, (d) teleconsultation, (e) telepathology, (f) teleradiology, and (g) remote assistance and patient management [4,5]. Telemedicine offers some fundamental advantages, such as (a) improved access to healthcare services for patients who live in distant locations or who have difficulty with mobility; (b) cost-effectiveness, reducing the number and cost of hospitalizations; and (c) improved quality of healthcare, if delivered in association with in-person services [6]. One of the main fields of telemedicine is telemonitoring, i.e., the wireless monitoring of physiological parameters in real time or using store-and-forward technologies [7]. Indeed, telemonitoring can be classified into synchronous and asynchronous. In the first case, the doctor can see the measurements live and take an immediate decision, while in the second case, clinical response is deferred [8]. Since the early 2000s, several publications have investigated the home monitoring of patients with respiratory diseases [9], congestive heart failure and arterial hypertension [10], and psychiatric and other chronic illnesses [11,12]. In addition, telemedicine has been used to assist in the home monitoring of blood glucose levels and pulmonary function [13,14].

To date, there is plenty of convincing evidence about the advantages of telemedicine also in terms of health economics [15]. In fact, previous studies have shown a reduced resource use, improved patient compliance and disease control, and reduced healthcare costs in chronic disease management [16]. In recent times, mobile health (mHealth) technologies have re-modernized telemedicine, especially in the cardiovascular field. mHealth technologies include smartphone applications, wearable devices, and handheld devices which can provide real-time monitoring of physiological measurements [17]. mHealth interventions have shown utility in the prevention, monitoring, and management of atrial fibrillation, heart failure, and myocardial infarction [17]. However, there is still little long-term data on the effect of mHealth interventions on death and hospitalization because most studies have focused on surrogate endpoints.

The COVID-19 pandemic has further increased awareness of the systems available for telemonitoring and encouraged the development of new systems [18,19,20]. This is mainly due to local pressure on hospitals and bed saturation, which have sometimes hindered the ability to cope with hospitalization demand [21,22]. In fact, telemonitoring offers the opportunity to carefully monitor patients with a confirmed or suspected case of COVID-19 from home and allows for the timely identification of worsening symptoms [23]. Additionally, it may decrease the number of hospital visits and admissions, thereby reducing the use of resources, optimizing health care capacity, and minimizing the risk of viral transmission [24]. In particular, mHealth devices have been used during the pandemic to monitor and identify potential patients and contain the COVID-19 outbreak [25]. For the most part, these devices were designed to measure one or few vital signs at a time. For example, a continuous body temperature monitoring program was developed to monitor temperatures of caregivers who may be exposed to COVID-19 [26]. However, identifying potential patients by temperature alone is neither sufficient nor accurate, because fever is also related to many other infections, and many patients are asymptomatic or do not experience a fever during the infection. Researchers have initiated app-based monitoring programs to detect and predict clinical progression using Fitbit and Apple watch wearables, which can extract heart rate and electrocardiography (ECG) curve [27]. Furthermore, respiratory rate is of critical importance to monitor the health status of COVID-19 patients, as a more severe lung involvement causes an increasing respiratory rate [28]. Compared with body temperature measurements, monitoring respiration fluctuations may serve as a more specific biomarker for COVID-19 prognosis, since most flu-like syndromes do not cause shortness of breath [29]. However, there is no device able to automatically measure respiratory rate to the best of our knowledge. Peripheral oxygen saturation (SpO_2_) measures the oxygen carrying capability of hemoglobin, and the normal level is around 94% to 100%. Respiratory involvement may cause hypoxemia, which makes monitoring pulse oximetry a crucial early warning factor to prevent lung diseases. Indeed, many pulse oximeters and wearable devices have been proposed to continuously and non-invasively monitor oxygen saturation levels at home [30].

In general, the home management of acute patients is challenging because of (a) the need to provide usual hospital care in an environment where multi-parameter monitoring is lacking, and (b) the ability to recognize clinical deterioration early on, that requires intensified care [31]. Telemonitoring provides remote and continuous measurement of vital parameters, allowing identification of patients who might benefit most from home visits or hospitalization [32]. However, as anticipated, most of the available devices consist of separate devices that measure individual vital parameters (e.g., oxygen saturation, blood pressure, body temperature), but there is no one device on the market that allows simultaneous measurement of all vital parameters. As each device requires specific skills to wear, as well as to measure and capture data, the average usability and usefulness of telemonitoring in real life is low due to poor patient compliance [33]. Indeed, the difficulty in device use has been demonstrated to be the most important factor related to telemedicine refusal by the users [34]. Therefore, there is a strong need for a simple and comprehensive tool that does not require medical knowledge or computer skills to use, and that can minimize the measurements needed while still providing reliable output data.

The Italian Startup VST (VST srl, Modena, Italy) developed a telemonitoring device, named ButterfLife, which provides a simple and fast simultaneous touch measurement of the five main parameters recommended by the World Health Organization (oxygen saturation, heart rate, respiratory rate, blood pressure, body temperature), ECG curve plots, and one-lead photoplethysmography (PPG) [35]. All data are made available to the doctor through a proprietary web platform. The system is based on artificial intelligence algorithms that are able to elaborate multiple output data from two analog input signals (ECG and PPG). While artificial intelligence is being increasingly implemented in e-Health devices, to our knowledge, ButterfLife is the only one provided with this technology (IppocraTech^®^), which has the advantage to be theoretically trained to infer other data from the same inputs [36,37]. This could be useful for the home care of patients suffering from acute or chronic diseases, allowing for timely recognition and management of clinical deterioration, a reduction in the need for hospitalization, and more convenient management of resources. We conducted a two-phase pilot study to evaluate the reliability, usability, and efficacy of the ButterfLife device both in the hospital and home settings.

## 2. Materials and Methods

This was a single-center pilot study conducted in two consecutive phases on two different populations. The primary objective (phase 1) was to evaluate the agreement between vital parameter measurements obtained with the ButterfLife device, versus those obtained by standard of care (SoC), in a group of patients hospitalized due to COVID-19 pneumonia. The secondary (phase 2) objectives were to evaluate the usability and effectiveness of the ButterfLife device in home telemonitoring of a group of polycomorbid patients. The study protocol was approved by the referral Ethics Committee (208-2022H) and it conformed to the Declaration of Helsinki and ICH E6 Guideline for good clinical practice. Each patient signed a written informed consent before enrollment.

The hardware used in this study was the ButterfLife electromedical system (CE certificate No. 0068/QPZ-DM/376-2021). Held between both hands with fingers and palms resting on the appropriate sensors for 90 s, the device acquired analog electrocardiography (ECG), photoplethysmography (PPG) and temperature signals from the sensors (Figure 1). The ECG and PPG signals have the following characteristics: (a) synchronous sampling with delay tolerance between the sampling instants of each individual ECG and PPG sample ≤2 ms; (b) sampling frequency equal to one of these values: 400 Hz, 512 Hz, 800 Hz, 1024 Hz; (c) minimum ECG spectral band: 0.05–100 Hz; (d) minimum PPG spectral band: 0.0–40 Hz; (e) quantization of ECG and PPG: 12 bit (or higher). ECG and PPG signals are conditioned with the following actions: (a) normalization in the scales [–1, 1] and [0, 1], respectively; and (b) filtering with infinite impulse response (IIR) filters. Room and body temperature signals are acquired in one or two moments (user’s choice) and present the following features: (a) at least two samples per second (512 Hz sampling rate was used); (b) a precision to a tenth of a degree and 12-bit resolution; (c) the accepted temperature range was from 0 to 45 °C. These signals were then processed by the stand-alone VSTFIVE software through specific algorithms based on biophysical analysis, multiple regressions, machine learning and neural networks. A specific user interface (https://my.vitalsignalstouch.com/) was incorporated into the software to manage signal loading, storage and display of calculation results. VSTFIVE is certified as a class IIa medical device (certificate 0068/QPZ-DM/184-2020 issued by the Notified Body MTIC Intercert Srl). Through specific measurement algorithms based on biophysical analysis, multiple regressions, machine learning and neural networks (covered by a Patent Cooperation Treaty patent pending), the VSTFIVE software integrates the digitalized signal inputs received from the sensors and provides the numerical measurement value of the above parameters. All input signals, except for temperature, underwent an automated quality control by the algorithms contained in VSTFIVE before being processed to the output measurements. Input signals which did not pass quality control were discarded and the corresponding output data were not calculated. After quality control, the VSTFIVE software integrated the signal inputs to provide the numerical measurement value of oxygen saturation, heart rate, respiratory rate, blood pressure and body temperature. The expected results every five seconds were: (a) one heart rate value obtained only from the ECG sensor; (b) one arterial oxygenation value calculated by the “ratio of ratios” method; (c) two blood pressure values, systolic and diastolic, respectively; (d) 1 respiratory rate value (the first value after thirty seconds, then every five seconds); and (e) two temperature values, body and environmental, respectively. These parameters, as well as one-lead electrocardiography and photoplethysmography raw curves, are made available to the users through a proprietary web platform with secure access (Figure 2).

Inclusion criteria were: (a) ability to understand and sign informed consent, and (b) aged 18 years or over. Exclusion criteria were: (a) dementia or decompensated psychiatric disorder; (b) quadriplegia/hemiplegia or quadriparesis/hemiparesis; and (c) any other condition that, in the opinion of the investigator, could significantly affect the patient’s ability to comply with the study interventions.

In the first phase of the study, 42 consecutive patients admitted to the high dependency respiratory ward of Trieste University Hospital due to COVID-19-related respiratory failure between January 2022 and March 2022 were recruited. After enrollment, each patient was asked to take a measurement with the ButterfLife device under the supervision of an investigator from the study team. Immediately thereafter, the investigator measured the same vital parameters with the standard of care used in the study setting: the Mindray BeneVision N17 (Mindray Medical Italia srl, Trezzano sul Naviglio, Italy) equipped with a blood pressure cuff, impedance sensor for respiratory rate, pulse oximeter for peripheral oxygen saturation and three-lead electrocardiograph for heart rate measurement; plus a Tyco Healthcare Genius 3 infrared tympanic thermometer for body temperature (Cardinal Health Inc., Dublin, Ireland).

In the second phase of the study, 8 chronic outpatients were recruited discretionally by the study team in March 2022. The patients were instructed in the use of the ButterfLife device, then asked to take a measurement at home every 24 h for 120 days. Patients were asked to report any new symptoms or the need for hospitalization, emergency room admission, or specialist visit to the principal investigator. The study team read all measurements daily and contacted the patient immediately if there were any major changes or deviations from normal parameters. Depending on the alteration found, the patient would be invited to repeat the measurement, prescribed second-level investigations, or provided an in-person visit. The primary outcome (phase 1) was concordance between vital parameters measured with ButterfLife and SoC. Secondary outcomes (phase 2) were: (a) number of measurements taken by each patient compared with the total number planned; (b) number of alterations in measured parameters that occurred recently; (c) avoidance of hospitalization; and (d) patient satisfaction assessed by a self-reported score (0 to 10) at the end of the study period.

Agreement between the measurements was assessed by visual inspection of Bland-Altman charts, and their correlation was assessed by Spearman’s Rho test. The mean difference between readings, upper and lower limits of agreement (95% confidence interval) were calculated from Bland–Altman output data. The sensitivity and specificity of the device in identifying the presence of pathological conditions were also calculated. Data are reported as mean (SD) or median (IQR) according to the distribution of variables.

## 3. Results

For the primary endpoint, all 42 patients were able to correctly perform the measurement with ButterfLife. The mean values (SD) of each parameter measured in the study population, as well as the mean (SD) of the difference between the individual parameters measured with ButterfLife and SoC for each patient and their correlation, are shown in Table 1. Graphical representation of the agreement between the measurements obtained with ButterfLife and SoC is also provided through Bland–Altman plots (Figure 3). The greatest discordance between ButterfLife and SoC, respectively, was observed for respiratory rate (mean difference −4.69 breaths per minute, limits of agreement 95% CI −5.91; −3.46). Spearman’s Rho test confirmed a significant correlation between measurements for all parameters except diastolic blood pressure (−0.18; *p* = 0.33) and oxygen saturation (0.20; *p* = 0.24).

Regarding the secondary endpoints, the baseline characteristics of the study population are shown in Table 2, and the results are shown in Table 3. The study population consisted of five women. Their age ranged from 52 to 91 years, with a median (IQR) of 72 (20.2). All patients performed at least one test per week, and the mean percentage of tests performed, compared with the number of planned tests, was 67.4%. Three patients performed <70% of scheduled tests due to (a) forgetfulness, two patients; or (b) lack of time, one patient. No patients reported difficulty with the device as a reason for poor compliance.

Daily reading of measured parameters enabled the study team to identify seven abnormalities (isolated extrasystoles, two patients; sinus tachycardia, one patient; previously unknown right bundle branch block, one patient; ventricular bigeminies, one patient; uncontrolled systolic hypertension, one patient; desaturation, one patient). Of these, six were confirmed during a subsequent in-person evaluation, while the right bundle branch block was not confirmed by a standard ECG. The patient who manifested ventricular bigeminies underwent a 24-h ECG halter and confirmatory cardiologic evaluation. The patient who had multiple episodes of mild systolic hypertension was successfully advised to increase antihypertensive therapy, while sporadic episodes of desaturation (SpO_2_ < 93%) were considered not clinically relevant by the pulmonologist, considering that the patient had chronic respiratory failure. Based on these data, the sensitivity of the device in identifying the onset of pathological conditions was 100% (95% CI 54.1% to 100%) and the specificity 88.9% (95% CI 51.8% to 99.7%).

No patients reported the appearance of new symptoms during the study period, nor did they require hospitalization or unplanned medical evaluations outside those indicated by the study team. Self-reported satisfaction reached a median (IQR) value of 9.5 (2.25). All patients were able to use the device correctly and independently, with no differences according to age. The mean percentage of valid tests to the total number of tests performed was 91.9%, but one patient had a lower percentage of valid tests (61.9%) due to parkinsonism-related tremor. There were no reports on device safety or problems with the hardware.

## 4. Discussion

In this study, we tested the reliability and usability of an all-in-one device, named ButterfLife, which allows for the contemporary monitoring of five vital parameters. The primary outcome was the agreement between measurements performed with ButterfLife vs. SoC in a group of forty-two patients affected by acute respiratory failure. The secondary outcomes were: (a) the proportion between effective and scheduled tests; (b) the number of newly occurred alterations; (c) the avoidance of hospitalization; and (d) patients’ satisfaction in a group of eight outpatients monitored for four months. The highest discordance between ButterfLife and SoC affected the respiratory rate (mean difference −4.69). Significant correlation was observed for all parameters except diastolic blood pressure and oxygen saturation (Spearman’s Rho −0.18 mmHg; *p* = 0.33 and 0.20%; *p* = 0.24, respectively). The mean proportion of tests performed was 67.4%, and no patient reported difficulties with the device. Seven conditions requiring medical attention were identified, with a sensitivity of 100% and a specificity of 88.9%. There were no hospitalizations in the study period, and the median patients’ satisfaction ranked as 9.5/10.

Multiparameter telemonitoring facilitates home care of patients, allowing early recognition of clinical deterioration and reducing the need for hospitalization. To the best of our knowledge, ButterfLife is the only system currently available for commercial use that simultaneously measures the five most important vital parameters. This results in increased usability for patients and safety for the physician who has to monitor several patients remotely. In our study, ButterfLife provided reliable measurements compared with standard of care. This was particularly remarkable because we performed the comparison on acute patients whose parameters were often unstable and divergent from normal. In fact, although the VSTFIVE software was implemented and fine-tuned with data derived from thousands of stable outpatients, it worked well even in our study conditions.

The overall usability of the device was high, with no differences among the elderly patients enrolled, and most patients enjoyed daily telemonitoring. ButterfLife was effective in highlighting various pathological conditions, enabling their management outside the hospital before patients experienced related symptoms.

Our study was a very preliminary feasibility trial, but it had several limitations.

First, it was designed in a predominantly descriptive manner. In fact, agreement between different readings of the same variables was evaluated through Bland–Alman plots, but the clinical relevance of the mean difference between each parameter may vary. The parameter that experienced the greatest disagreement between ButterfLife and SoC was respiratory rate (−4.69 mean difference, 95% limits of agreement −5.91 to −3.46). This finding is not surprising, considering that respiratory rate was estimated by impedance pneumography in SoC or by Pulse Transit Time analysis in ButterfLife, which are both indirect methods prone to possible errors. Indeed, although we have not found a direct comparison between these methods in the literature, the mean difference in RR assessed by direct observation of the chest wall movements vs. impedance pneumography varied between 0.4 ± 5.9 bpm, 95% limits of agreement −11.1 to 11.9 [38] and 1.7 ± 4.4, 95% limits of agreement −13.3 to 16.8 [39] between studies. However, ButterfLife tended to underestimate the respiratory rate in the whole spectrum of measurements, with a progressively increasing difference for higher values (Figure 3, panel D). This was consistent with previous data reporting on impedance pneumography vs. electrocardiography-derived measurement of respiratory rate [40].

Another limitation relates to the small study population and discretionary patient enrollment which may involve selection biases. In fact, we enrolled as many patients as possible based on the availability of the devices, and we tried to include patients of different ages and sexes and suffering from different diseases.

Furthermore, a considerable number of patients (3/8) fell below the average threshold of 67.4% of actual versus planned measurements. However, this was not due to difficulties in using the device and was in disagreement with the high overall satisfaction score expressed by the patients themselves.

One patient had a large number of invalid tests due to the low quality of the input signals, which generally depends on (a) poor grip of the hands on the sensors, (b) dirty hands, (c) peripheral hypoperfusion, or (d) excessive movement during the measurement. While this may appear to be a weak point of the system, it should be emphasized that automated quality control is essential to ensure maximum measurement reliability.

If larger studies confirm our findings, the use of ButterfLife could be extended to a wide range of clinical settings and clinical conditions, allowing both cost savings and time savings through early disease detection, reduced hospital admissions and possible follow-up of patients living in isolated areas with difficulty in accessing hospital care. The same device could find application in contexts other than the hospital, where medical advice is discontinuous and can be centralized (e.g., nursing homes, insurance companies, general practitioners, workplaces, public areas and services, etc.), allowing a greater cost-effective management of resources. In the near future, the device could be trained to theoretically deduce any laboratory parameters from the same input signals, without the need for venous sampling, thanks to further refinement of artificial intelligence algorithms which are being increasingly implemented in e-Health technologies [36,37]. For example, non-invasive detection of blood glucose and lactate levels is currently under development.

## 5. Conclusions

In conclusion, this is the first study that reports the reliability and feasibility of multi-parameter home telemonitoring detected simultaneously with a single device, which can represent a turning point for the home management of people suffering from both acute and chronic diseases. The system was usable and largely accurate for the measurement of oxygen saturation, heart rate, blood pressure and body temperature, but it tended to underestimate respiratory rate compared with standard of care. ButterfLife proved to be a reliable and easy-to-use device capable of simultaneously assessing multiple vital parameters, in addition to providing a single-lead ECG trace and photoplethysmogram, in both the hospital and the home setting. Further randomized controlled trials are needed to evaluate the clinical benefits of using ButterfLife in telemedicine programs.

## Figures and Tables

**Figure 1 diagnostics-12-03115-f001:**
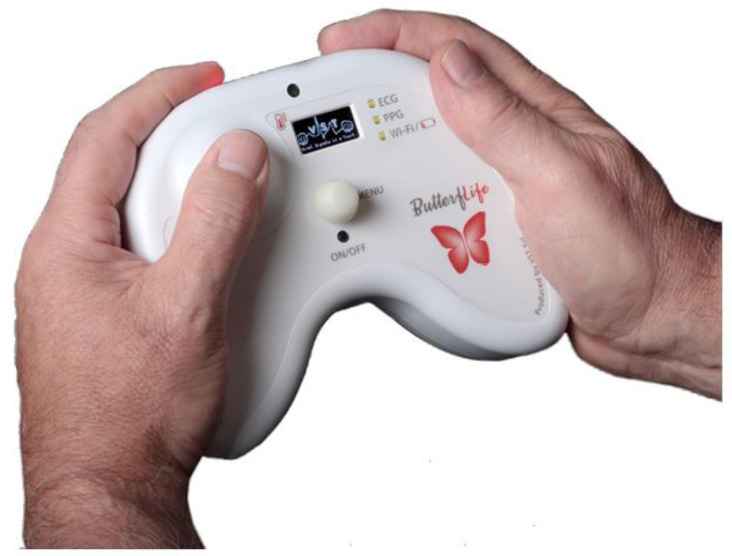
The ButterfLife device. In order to perform a test, the device needs to be held between both hands for 90 s with palms placed on the lateral electrocardiography sensors and fingers placed on the photoplethysmography sensors.

**Figure 2 diagnostics-12-03115-f002:**
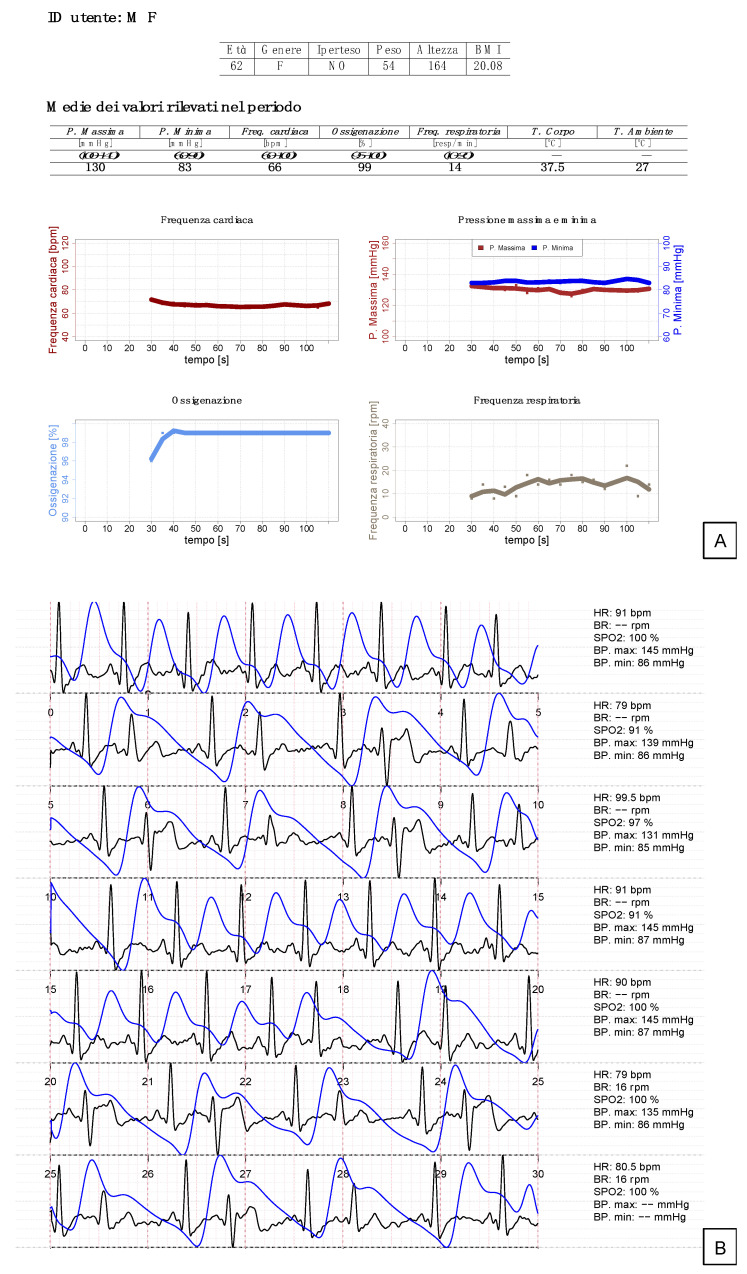
Output measurements as they appear in the report available in real-time for both the patient and the doctor through the proprietary web platform. (**A**): table above, mean values for each parameter obtained during the 90-s test; lower graphs, graphical representation of the time-course of each parameter during the 90-s test. (**B**): one-lead electrocardiography plot and superimposed photoplethysmography curve (blue). On the right are the numerical values of each parameter every 5 s. Missing data: Measurements not displayed due to failed internal quality control.

**Figure 3 diagnostics-12-03115-f003:**
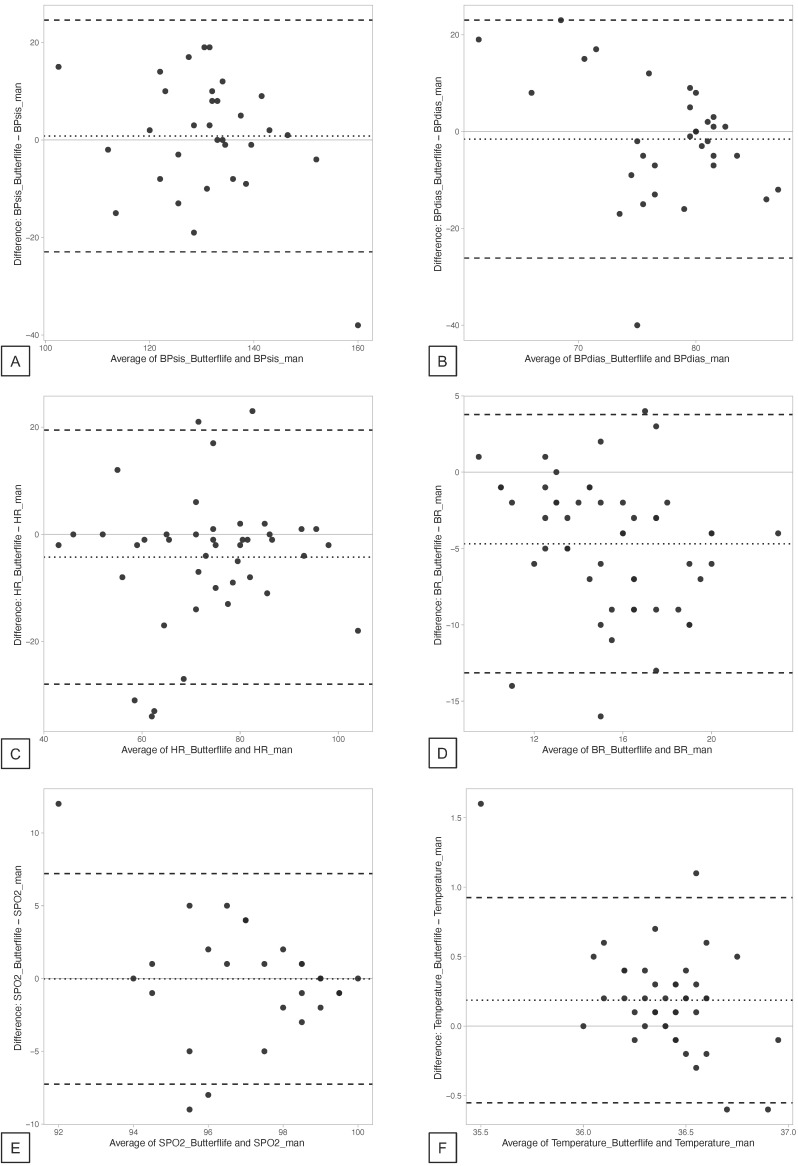
Bland–Altman plots of the single measurements ((**A**), systolic blood pressure; (**B**), diastolic blood pressure; (**C**), heart rate; (**D**), respiratory rate; (**E**), oxygen saturation; (**F**), body temperature) obtained with ButterfLife vs. SoC. X-axis: average of the measurements obtained with ButterfLife and SoC; Y-axis: difference between measurements obtained with ButterfLife and SoC. The dotted line indicates the mean difference between measurements, while the dashed lines indicate the upper and lower 95% limits of agreement.

**Table 1 diagnostics-12-03115-t001:** Simultaneous measurement of vital parameters with ButterfLife vs. SoC.

	ButterfLife, Mean (SD)	SoC, Mean (SD)	ΔButterfLife—SoC, Mean (Limits of Agreement 95% CI) *	Correlation ^¶^	*p*-Value ^¶^
SBP (mmHg)	131.72 (11.03)	130.91 (14.31)	0.81 (−3.63; 5.25)	0.54	<0.01
DBP (mmHg)	76.75 (6.55)	78.31 (9.87)	−1.56 (−6.15; 3.03)	−0.18	0.33
HR (bpm)	71.56 (15.47)	75.81 (14.18)	−4.26 (−8.02; −0.49)	0.65	<0.01
RR (bpm)	13.12 (3.18)	17.80 (4.02)	−4.69 (−5.91; −3.46)	0.33	0.02
SpO_2_ (%)	97.58 (2.27)	97.61 (3.04)	−0.03 (−1.29; 1.24)	0.20	0.24
Temperature (°C)	36.50 (0.71)	36.25 (0.46)	0.19 (0.07; 0.30)	0.36	0.02

* Data from Bland–Altman plots. ^¶^ Data from Spearman’s Rho correlation test. Legend: ΔButterfLife-SoC, mean of the differences of the measurements obtained with ButterfLife and SoC; SoC, standard of care; SBP, systolic blood pressure; DBP, diastolic blood pressure; HR, heart rate; RR, respiratory rate; SpO_2_, peripheral oxygen saturation; bpm, beats/breaths per minute. Standard care: BP, HR, RR, SpO_2_: Mindray BeneVision N17. Temperature: Tyco Healthcare Genius 3 infrared tympanic thermometer.

**Table 2 diagnostics-12-03115-t002:** Baseline characteristics of the study population.

Patient								
	1	2	3	4	5	6	7	8
Age	91	75	53	62	69	79	52	83
Sex	F	M	M	F	M	F	F	F
BMI	22.2	27.4	23.3	20.1	31.6	22.0	22.8	30.4
Smoker	No	Yes	No	Yes	No	No	No	Former
Main disease	CRF (IPF) Type 2 DM CHF	Hypertension; Type 2 DM	Hypertension	Hypertension	Hypertension; Lung cancer Parkinsonism	CHF; MGUS	Asthma	CRF (COPD); kyphoscoliosis
Oxygen therapy	Yes	No	No	No	No	No	No	Yes
Baseline ECG abnormalities	None	None	None	None	None	None	None	None

Legend: BMI, body mass index; IPF, idiopathic pulmonary fibrosis; DM, diabetes mellitus; COPD, chronic obstructive pulmonary disease; CRF, chronic respiratory failure; CHF, chronic heart failure; MGUS, monoclonal gammopathy of unknown significance.

**Table 3 diagnostics-12-03115-t003:** Secondary endpoints.

Patient								
	1	2	3	4	5	6	7	8
No. Tests performed (% of test due)	107 (89.2)	21 (17.5)	123 (102.5)	46 (38.3)	126 (105.0)	86 (71.7)	20 (16.7)	118 (98.3)
No. of valid tests (% of tests performed)	104 (97.2)	20 (95.2)	118 (95.9)	42 (91.3)	78 (61.9)	85 (98.8)	20 (100.0)	112 (94.9)
Identified condition	None	None	Sinus tachycardia	Bigeminism	Hypertension	Isolated extrasystoles	RBBB; Isolated extrasystoles	Desaturation
Confirmed condition	None	None	Sinus tachycardia	Bigeminism	Hypertension	Isolated extrasystoles	Isolated extrasystoles	Desaturation
Need for hospitalization	No	No	No	No	No	No	No	No
Patient’s satisfaction (0–10)	10	8	10	10	7	9	7	10
Problems with the device	None	None	None	None	Parkinsonism	None	None	None
SBP, mmHg (median, IQR)	132.0 (5.8)	139.0 (3.0)	123.0 (7.0)	136.0 (8.0)	146 (4.0)	132.0 (3.0)	127.0 (3.0)	133.0 (6.0)
DBP, mmHg (median, IQR)	82.0 (1.0)	81.0 (0.2)	80.0 (3.0)	83.0 (1.0)	81.0 (2.5)	83.0 (0.0)	73.0 (2.0)	81.0 (1.0)
HR, bpm (median, IQR)	77.0 (7.5)	69.0 (7.0)	68.0 (12.0)	76.0 (15.2)	71.0 (6.2)	67.0 (5.2)	77.0 (8.2)	64.0 (5.5)
RR, bpm (median, IQR)	15.5 (5.0)	15.0 (2.0)	12.0 (4.0)	14.0 (3.0)	12.0 (4.0)	15.0 (2.5)	13.0 (6.0)	13.0 (4.0)
SpO_2_, % (median, IQR)	99.0 (0.0)	99.0 (0.0)	99.0 (2.0)	99.0 (1.2)	98.0 (0.5)	96.0 (5.0)	98.0 (2.5)	94.0 (3.0)
Temperature, °C (median, IQR)	36.3 (0.0)	36.3 (0.0)	36.5 (0.2)	36.5 (0.7)	36.2 (0.2)	36.4 (0.2)	36.6 (0.0)	36.4 (0.5)

Abbreviations: RBBB, right bundle branch block; IQR, interquartile range; SBP, systolic blood pressure; DBP, diastolic blood pressure; HR, heart rate; RR, respiratory rate; SpO_2_, peripheral oxygen saturation; bpm, beats/breaths per minute.

## Data Availability

De-identified participant data will be made available upon motivated request to the Corresponding Author. The proposed use of the data and analyses must be approved by the Scientific Committee.

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
