# Peer review of "Clinical Evaluation of the ButterfLife Device for Simultaneous Multiparameter Telemonitoring in Hospital and Home Settings"

_diagnostics, 2022, doi:10.3390/diagnostics12123115_

Round 1

Reviewer 1 Report

The study was scientifically sound. Consecutive patients were recruited and the reference device was sound. Daily data collection for a suitable duration ensured clear signal. I also felt the writing emphasized the correct points - drawing out what is perhaps the main finding: that the system was usable and largely accurate - expect for RR. 

My question is really around RR: "which was deemed to be clinically acceptable". I do not understand how the clinicians came to this conclusion given the fact that the domain was the respiratory ward. I imagine a more suitable test would have been to pre-define the acceptable accuracy level or to cite a study that found the clinically meaningful error in RR. Given the mean RR was 17.8, an error of 26% seems clinically significant to me. Perhaps the clinical team did not trust the reference device. 

I think expounding on this may be useful to readers: Why did the clinicians deem this level of error to be clinically acceptable?

Also: it would be good to understand if the errors were at the higher ends of the RR spectrum? Or at all ends? We have done these studies before and it is useful to find the range of RR values for which the system is accurate.

I commend the authors for a robust study with encouraging results and a well-written paper.

Reviewer 2 Report

no cpecific comments

Author Response

We would like to thank the Reviewer for having taken part to the review process of our paper.

Reviewer 3 Report

Dear Authores,

The paper showed the studies in context of practical and current issues in mHealth. The aim is to show the evaluation of the ButterfLife device.

Please describe the knowledge gap of mHealth studies and the aim of your paper in the introduction, and add the research methodology with methods for appropriate research aims.

Additionally describe the limitation, and consider to compare your research findings to previous studies in the research field. Please show the background of the study with the appropriate references of the research field of artificial intelligency in health care; the context and the scope of the researh need to be added. 

Best regards

Round 2

Reviewer 3 Report

Dear Authors,

Thank you for your answers.

Please add the knowledge gap, and knowledge implications of your research. 

Additionally, instead of practical issues, the theoretical background of study should be described.

Best regards
